# Opening the Black Box:
# Automated Software Analysis for Algorithm Selection

Damir Pulatov[1]  Marie Anastacio[2]  Lars Kotthoff[1]  Holger Hoos[2,3]

[1]University of Wyoming
[2]Leiden University
[3]RWTH Aachen

**Abstract**  Impressive performance improvements have been achieved in many areas of AI by meta-algorithmic techniques, such as automated algorithm selection and configuration. However, existing techniques treat the target algorithms they are applied to as black boxes – nothing is known about their inner workings. This allows meta-algorithmic techniques to be used broadly, but leaves untapped potential performance improvements enabled by information gained from a deeper analysis of the target algorithms. In this paper, we open the black box without sacrificing universal applicability of meta-algorithmic techniques by automatically analyzing algorithms. We show how to use this information to perform algorithm selection, and demonstrate improved performance compared to previous approaches that treat algorithms as black boxes.

## 1 Introduction

The past decade has seen the rise of meta-algorithmic techniques that enable the more efficient and intelligent use of existing algorithms, alleviating some of the challenges encountered in the design of new algorithms. This has given rise to the field of Automated Machine Learning [Hutter et al., 2019], which heavily relies on meta-algorithmic techniques. A prominent example of the success and impact of meta-algorithmic techniques is algorithm selection, where we choose from a given portfolio of algorithms the one likely to perform best on a given problem instance.

The *per-instance algorithm selection problem*, in which, given a problem instance, one of several available algorithms is to be chosen to solve that instance most effectively and efficiently, has been first considered in the 1970s [Rice, 1976]. The first prominent system that demonstrated the promise of algorithm selection was SATzilla [Xu et al., 2008], which chooses the most suitable algorithm from a portfolio of satisfiability (SAT) solvers. SATzilla showed excellent performance (including several gold medals) in SAT solver competitions. Since then, many more examples have shown the promise of algorithm selection, where we choose from among a set of existing algorithms, and algorithm configuration, where we tune the parameters of algorithms to achieve the best performance. Among them are 3S [Kadioglu et al., 2011], Proteus [Hurley et al., 2014], SMAC [Hutter et al., 2011] (used in e.g. auto-sklearn [Feurer et al., 2015]) and TPOT [Olson et al., 2016]. The growing interest in per-instance algorithm selection over the past decade has led to the establishment of the Algorithm Selection Benchmark Library, ASlib [Bischl et al., 2016], and to two competitions for algorithm selection systems [Lindauer et al., 2019].

While these systems are based on different techniques and leverage different ideas and assumptions, they share a common trait – the algorithm(s) they leverage are treated as black boxes; that is, we know nothing about how the algorithms work internally. Instead, we gather information by probing given algorithms repeatedly, usually by observing their performance in solving a set of problem instances. In this work, we argue that by ignoring information that can be obtained through the automatic analysis of algorithms, we leave untapped useful knowledge that can substantially improve the performance of meta-algorithmic approaches. We open the black box without

compromising the general applicability of meta-algorithmic techniques by automatically analyzing the algorithms under consideration. We demonstrate that considerable performance improvements can be achieved using this additional information for algorithm selection.

The performance improvements achieved by our approach result in improved efficiencies, thus reducing the amount of computation necessary to solve a problem. This reduction in computation can also contribute to a more sustainable and energy-efficient AI. We have not identified any potential negative societal impacts in either our work or possible applications of our work.

## 2 Related Work

Choosing the most promising approach for solving a given problem has always been of interest, in particular for hard combinatorial problems (e.g. [Allen and Minton, 1996, Weerawarana et al., 1996, Lobjois and Lemaître, 1998]). The idea of using portfolios for this purpose was introduced by [Huberman et al., 1997] and popularized in AI by [Gomes and Selman, 2001]. These early approaches that leverage complementarity in the performance characteristics of algorithms did not perform algorithm selection as we consider it here, but rely heavily on parallel processing to be able to run several or all portfolio algorithms concurrently. They were soon followed by approaches performing algorithm selection in the sense we consider here (e.g. [Guerri and Milano, 2004, Beck and Freuder, 2004, Hough and Williams, 2006]).

However, it took several decades before algorithm selection procedures achieved significant performance improvements in practical settings. One of the first such systems is SATzilla [Leyton-Brown et al., 2003, Xu et al., 2008], which solves the prominent SAT problem using a rich set of instance features and draws from a broad set of high-performance SAT solvers. Different versions of SATzilla won several SAT competitions and prompted the development of further algorithm selection systems for SAT, e.g. [Roussel, 2012, Malitsky et al., 2013].

In recent years, algorithm selection systems have gained prominence in many application domains, including ASP [Hoos et al., 2014], CSP [Hurley et al., 2014], QBF [Pulina and Tacchella, 2009], AI planning [Vallati, 2012], TSP [Kerschke et al., 2018], and operations research [Tierney and Malitsky, 2015]. The growing interest in per-instance algorithm selection over the past decade has led to the establishment of the Algorithm Selection Benchmark Library, ASlib [Bischl et al., 2016], and to two competitions for algorithm selection systems [Lindauer et al., 2019]. We refer the interested reader to two surveys on algorithm selection for additional details and an overview of further work in the area [Kotthoff, 2014, Kerschke et al., 2019].

Per-instance algorithm selection systems usually apply machine learning to determine which algorithm to run on a given problem instance, based on expert-defined, cheaply computable features of problem instances. These features can either be used to directly predict the best-performing algorithm (e.g. [Kadioglu et al., 2010, Gent et al., 2010, Pfahringer et al., 2000]), or to obtain performance predictions for all given algorithms and choose the algorithm with the best predicted performance; the latter approach was used in the original SATzilla system.

Recent work has applied deep learning techniques to obviate the need for human experts to explicitly define problem instance features [Loreggia et al., 2016]. While this initial study showed promising results, to the best of our knowledge, the approach has not been developed further. Another line of work has applied deep learning more generally to directly learn algorithms and when to use them [Khalil et al., 2016, Khalil et al., 2017, Dai et al., 2017]. As most deep learning approaches, they require significant amounts of data and resources to train models compared to other approaches to algorithm selection though.

While there has been work on taking into the account additional features from the algorithms, as shown in [Tornede et al., 2020], this approach is much more limited than what we present here and requires an "extremely large" set of algorithms. To the best of our knowledge, is only the case in this particular paper. The algorithm features are latent, similar to recommender systems (as pointed out in the paper) and have no intrinsic meaning, as opposed to our algorithm features.

Additionally, representing an algorithm by its hyperparameters is a well-established way of taking information on the algorithm into account in the meta-algorithmic process (e.g. [Hutter et al., 2006]), but it requires expert knowledge – the developer must decide which hyperparameters and value ranges to expose. In contrast, our approach does not require this and is completely automated.

Finally, [Adriaensen and Nowé, 2016] make a general point on opening up the "black box" in meta-algorithmics, in particular in the context of automated algorithm design. The paper proposed, among other things, to analyze the source code of algorithms in order to achieve their goal, but the authors did not present any results.

We open the black box of the algorithms that are being modeled. To the best of our knowledge, there are no existing techniques that allow to automatically analyze the inner workings of algorithms to facilitate per-instance algorithm selection.

## 3 Methodology

In many cases, the source code of an algorithm is available. The software engineering community has developed ways of quantifying the complexity and other properties of source code for decades – we can leverage their work for our purposes. Such source code features can be computed quickly by readily-available tools for many different programming languages.

Our second source of information is the abstract syntax tree (AST) of the source code of an algorithm that is constructed as part of the compilation process. It provides a higher-level view of the algorithm and makes connections between its individual components explicit. Moreover, the resulting graph structure can be analyzed and quantified using standard tools for the analysis of graphs. Again, the tools are readily available and most features can be computed quickly.

Formally, following [Bischl et al., 2016], we have a set $\mathcal{I}$ of problem instances drawn from a distribution $\mathcal{D}$, a space of algorithms $\mathcal{A}$, a vector of instance features $\mathcal{F}_\mathcal{I}$ of arity $\|\mathcal{I}\|$, a vector of algorithmic features $\mathcal{F}_\mathcal{A}$ of arity $\|\mathcal{A}\|$, and a performance measure $m : \mathcal{I} \times \mathcal{A} \to \mathbb{R}$. The per-instance algorithm selection problem with algorithm features is to find a mapping $s : \mathcal{F}_\mathcal{I} \times \mathcal{F}_\mathcal{A} \to \mathcal{A}$ that optimizes $\mathbb{E}_{i \sim D} m(i, s(i))$, i.e., the expected value of the performance measure for instances $i$ distributed according to $\mathcal{D}$, achieved by running the selected algorithm $s(i)$ for instances $i$. The difference to the original definition in [Bischl et al., 2016] is that the mapping includes both instance ($\mathcal{F}_\mathcal{I}$) and algorithmic features ($\mathcal{F}_\mathcal{A}$).

Following the definitions above, the single best algorithm, or solver, is defined as the solver that has the best performance averaged across all instances, while the virtual best solver is a solver that perfectly selects the best solver from $\mathcal{A}$ on a per-instance basis. More formally, the virtual best solver is defined as $\{arg\,min_a \mathbb{E}_{i \sim D} m(i, s(i)) \mid i \in \mathcal{I}, a \in \mathcal{A}\}$ where $i$ and $a$ are individual instances and algorithms that come from the set of instances and algorithms, $\mathcal{I}$ and $\mathcal{A}$, respectively. Similarly, the single best solver is defined as $arg\,min_a \frac{\sum_{i \in \mathcal{I}} \mathbb{E}_{i \sim D} m(i, s(i))}{\|\mathcal{I}\|}$.

In addition to identifying sources of information about an algorithm, we need to develop ways of taking this information into account during performance modeling and algorithm selection. Current state-of-the-art approaches to empirical performance modeling build one model per algorithm and predict its performance based on the features of a problem instance to solve [Hutter et al., 2014]. Simply adding the features of an algorithm to such models will not improve performance, as the features will be constant for a given model and algorithm. However, the reason for building individual models for each algorithm is that otherwise there is no way to distinguish between them. Our proposed algorithm features provide a way to do exactly that, obviating the need for multiple performance models – we can build a single, unified model for all algorithms.

One of the advantages to this approach is that the single model takes much more information into account than individual models, making it possible, for example, to learn relationships between algorithms that can improve the quality of performance predictions. Note that algorithm features

do not only permit to distinguish between different algorithms, but also quantifying how similar they are – further useful information that a combined performance model can take into account.

We also consider models that predict the performance differences between pairs of algorithms and take the difference between the features of the respective algorithms in addition to the instance features into acocunt. Again, we are able to use just a single unified model for an entire portfolio of algorithms.

## 4 Empirical Evaluation

To evaluate the promise of our approach, we leveraged the widely used ASlib benchmark library [Bischl et al., 2016], which provides algorithm selection scenarios from many different application domains. All experiments were run on compute nodes with Intel Broadwell CPUs with 32 cores clocked at 2.1 GHz, 40 MB of CPU cache, and 128 GB of RAM, running Red Hat Enterprise Linux 7.7.

### 4.1 Algorithm Selection Scenarios

As we require the source code for the algorithms to be available, we limited our empirical study to ASlib scenarios that satisfy this constraint. Further, we created additional scenarios for our evaluation and contributed these to ASlib. Table 1 gives a summary of the scenarios we used. These scenarios represent a variety of different applications of algorithm selection. Starred ASlib scenarios were modified to exclude algorithms for various reasons such as lack of available source code in the submission, lack of support for a particular language in our tools, etc.

For further details on how scenarios were modified, we refer the interested reader to Appendix A.

| Scenario | Instances | Algorithms |
|---|---|---|
| SAT03-16_INDU* | 2000 | 8 |
| SAT11-INDU | 353 | 18 |
| SAT11-RAND* | 600 | 8 |
| SAT11-HAND* | 296 | 11 |
| GRAPHS-2015* | 5725 | 4 |
| TSP-LION2015 | 3106 | 4 |
| SAT18-EXP | 353 | 37 |
| GLUHACK-18 | 353 | 8 |
| MAXSAT19-UCMS | 572 | 7 |

Table 1: Overview of scenarios used in this paper with the number of problem instances and algorithms. ASlib scenarios are at the top and our new scenarios at the bottom. Starred ASlib scenarios were modified to exclude algorithms.

### 4.2 Algorithm Features

For each algorithm, we extracted both source code and AST features. Table 2 summarizes the algorithm features we use to evaluate the promise of our approach.

Our *code features* were computed using an open-source tool [Metrix++, 2019]. We computed cyclomatic complexity, maxindent complexity, number of lines of code, and size in bytes, for the entire source code of a given algorithm and the average values across different regions of the code. The resulting feature set is computed within a few seconds for each of our algorithms.

Our *AST features* are based on the abstract syntax tree obtained during compilation of a given algorithm; we generated the ASTs using the open-source compiler Clang [Clang, 2019] and collected

the proportion of nodes of each type (based on the Clang AST types); of edges that link each pair of types; and of each data-types to which the operators nodes are applied. To generate graph features from the AST, we used the NetworkX library [NetworkX, 2019]. We computed the number of nodes and edges; degrees of the nodes; transitivity; clustering coefficient [Fagiolo, 2007]; and depth. For degree and clustering coefficient, we compute minimum, maximum, mean, and variance across all nodes. For depth, we compute these measures across all leaves. We also compute the entropy of the distributions for degree and depth across the entire tree and the leaf nodes, respectively. The clustering coefficient is often zero, for which the entropy cannot be computed. Computation of the AST features takes from a few seconds up to hours per algorithm, but is a one-time expense.

To investigate whether the algorithm features serve a purpose beyond purely distinguishing between different algorithms, i.e. to assess to which degree they are useful for algorithm selection for a given problem instance, we include a third set containing only a single *dummy feature* that represents a unique identifier of the algorithm.

| Type | Name | Explanation | # Features |
|------|------|-------------|------------|
| Code | Lines of code | | 2 |
| | Cyclomatic complexity | number of independent execution paths [McCabe, 1976] | 2 |
| | Maxindent complexity | maximum level of indentation [Tornhill, 2018] | 2 |
| | Size of the sources | | 2 |
| | Number of files | | 1 |
| AST | Node count | number of nodes in the AST | 1 |
| | Edge count | number of edges in the AST | 1 |
| | Degrees of the nodes | | 5 |
| | Transitivity | number of triangles on three connected nodes | 1 |
| | Clustering coefficient | measure the local connectivity of a node [Fagiolo, 2007] | 4 |
| | Depth | distance from the root node to each leaf | 5 |
| | Node type | based on Clang AST | 6 |
| | Edge type transition | based on Clang AST | 36 |
| | Operation type | data types operators are applied to | 7 |
| Dummy | ID | identifier of algorithm | 1 per algorithm |

Table 2: Algorithm features considered in our study, grouped by type.

### 4.3 Algorithm Selection

There are different approaches to algorithm selection; we focus on the two methods described in Section 3 – models to predict the performance of each individual algorithm and models to predict the performance difference between pairs of algorithms. These models take our new algorithm features for individual algorithms and the difference of those features for pairs of algorithms into account. Dummy identifier features are encoded in a one-hot way with a one indicating that a particular algorithm or pair of algorithms was measured, with features for all other algorithms or pairs of algorithms set to zero.

Each ASlib scenario defines 10 cross-validation folds to ensure repeatable and robust performance evaluation. The performance models are trained on the data from nine folds and their

performance is evaluated on the remaining fold. This is repeated for each combination of training and testing folds, for a total of ten iterations. We report the average performance across all folds and the standard deviation. We use random forest regression models to predict the performance of each algorithm and imputed missing instance feature values in all scenarios by the mean value across all non-missing values for numeric features, and NA for categorical features, following [Bischl et al., 2016].

We performed basic hyperparameter optimization of the random forest models. Specifically, we varied the number of trees in the random forest from 10 to 200, and the number of features used at each split when growing trees from 1 to 30. The hyperparameters and ranges are the same as in those used by [Bischl et al., 2016]. We performed a random search in this parameter space with 250 samples and a nested cross validation with ten outer and three inner folds, selecting the model with the lowest average PAR10 across the outer cross-validation folds for our comparison.

We consider two metrics for evaluating the performance of algorithm selection systems – misclassification penalty (MCP) and penalized average running time with a factor of 10 (PAR10). The misclassification penalty is defined as the additional cost incurred because a sub-optimal algorithm was chosen, i.e. additional running time or decreased accuracy of a given machine learning model. PAR10 is defined as the average running time over a given set of problem instances, with timeouts counted as ten times a user-specified cutoff time.

We compare to the single and virtual best solvers, which provide bounds on the performance of an algorithm selection system – it cannot be better than the virtual best solver (VBS) and should not be worse than the single best solver (SBS), as there is no performance gain from algorithm selection otherwise. We normalize the performance of an algorithm selection system to the fraction of the gap closed between single and virtual best (see e.g. [Bischl et al., 2016]). This allows to directly compare performance across different scenarios.

Due to the size of the scenario and the amount of computation required, we were unable to consider all algorithms for SAT18-EXP when using models that predict performance difference. Therefore we modified SAT18-EXP to exclude seven solvers. However all other scenarios were used without modifications, and the full SAT18-EXP scenario was used for experiments where the performance of each algorithm is predicted directly and a selection is made based on this predicted performance. For further details on what solvers were excluded and the motivation behind this decision can be found in Appendix B.

## 5 Results

We now present the results of our experiments, demonstrating the promise of our algorithm features and the performance improvements that can be achieved.

### 5.1 Algorithm Selection

Figure 1 shows the results of our experiments for algorithm selection in terms of the MCP metric. Adding solver features improves algorithm selection for the majority of the scenarios we considered; 8 out of 9 scenarios for regression models and 6 out of 9 for pairwise regression models with an additional scenario resulting in a tie. While adding algorithm features does not help for all scenarios, we do see substantial performance improvements in some cases, with improvements of up to 26% for regression models and 28% for pairwise regression models.

For the TSP-LION2015 scenario, adding solver features improves algorithm selection performance to the point where it is better than the single best solver, closing an additional 26% between single and virtual best solver in terms of MCP for the regression model and 28% for the pairwise regression model. For GRAPHS-2015, we close an additional 15% of the gap for regression and 7% for pairwise regression. For SAT11-INDU, we close an additional 10% of the gap to the virtual best solver for regression and a more modest 6% of the gap for the pairwise regression model. We also

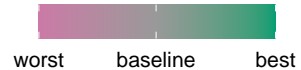

worst   baseline   best

|  | Regression | | | | | Pairwise Regression | | | | |
|---|---|---|---|---|---|---|---|---|---|---|
|  | I | II | IC | IA | ICA | I | II | IC | IA | ICA |
| TSP–LION2015 | −19 ±29.41 | −36 ±35.34 | −13 ±29.21 | **7 ±13.38** | 5 ±9.4 | −29 ±21.68 | −29 ±31.9 | −12 ±28.34 | −3 ±9.11 | **−1 ±4.85** |
| SAT18–EXP | 61 ±20.08 | 62 ±19.48 | 57 ±17.35 | 62 ±22.01 | **63 ±19.64** | **59 ±7.02** | 34 ±16.21 | **59 ±4.53** | 47 ±14.59 | 46 ±11.78 |
| SAT11–RAND* | **91 ±6.8** | 90 ±7.63 | 88 ±10.1 | 89 ±8.41 | 90 ±5.92 | **92 ±5.42** | **92 ±5.41** | 91 ±6.06 | 82 ±11.63 | 83 ±10.25 |
| SAT11–INDU | 31 ±39.64 | 34 ±29.84 | **41 ±19.78** | 34 ±23.71 | 36 ±24.67 | 39 ±26.65 | 34 ±36.34 | **45 ±34.73** | 43 ±31.89 | 35 ±36.44 |
| SAT11–HAND* | 42 ±53.59 | 42 ±41.22 | 43 ±29.73 | 45 ±35.26 | **51 ±44.75** | 45 ±45.64 | 46 ±52.59 | 42 ±37.9 | 50 ±41.51 | **51 ±22.81** |
| SAT03–16_INDU* | 46 ±19.46 | 49 ±9.73 | 48 ±16.33 | **51 ±16.65** | 47 ±16.9 | **52 ±18.43** | 41 ±20.41 | 51 ±15.08 | 40 ±15.91 | 44 ±12.61 |
| MAXSAT19–UCMS | 36 ±22.4 | 34 ±35.38 | **45 ±21.86** | 40 ±26.93 | 34 ±28.04 | 41 ±30.35 | 47 ±33.36 | **51 ±34.14** | 21 ±30.3 | 26 ±38.24 |
| GRAPHS–2015* | 54 ±49.34 | 53 ±78.76 | **69 ±70.84** | 52 ±76.94 | 52 ±181.36 | 56 ±113.83 | 62 ±176.06 | 54 ±92.41 | **63 ±72.86** | 58 ±65.11 |
| GLUHACK–18 | 40 ±27.54 | 39 ±35.83 | **48 ±31.76** | 44 ±41.61 | **48 ±30.03** | 40 ±31.25 | 42 ±36.92 | **47 ±18.91** | 40 ±28.49 | 45 ±27.43 |

Figure 1: Impact of algorithm features on the performance of algorithm selection based on regression and pairwise regression models. The top value in each cell is the average percentage value of the gap closed between single best (0) and virtual best solver (100) in in terms of misclassification penalty (MCP) across cross-validation folds; the bottom value is the standard deviation. Values greater than zero indicate that performance is better than the single best solver, negative numbers indicate that it is worse. Average values have been rounded to integers and standard deviations to two decimal places. Starred ASlib scenarios were modified to exclude some algorithms. The two columns on the left show the baselines to which we compare; instance features only (I) and instance and dummy ID features (II). The algorithm feature sets we consider are instance and code features (IC), instance and AST features (IA), and instance, code, and AST features (ICA). The best value for a particular scenario is shown in **bold**.

see impressive results in MAXSAT19-UCMS, where we close an additional 9% and 10% of the gap for regression and pairwise regression, respectively. For other scenarios where we improve, the performance difference is smaller, from 2% to 9% for regression and from 6% to 7% for pairwise regression. We note that algorithm selection performance improves by several percentage points on the GLUHACK-18 scenario, where all algorithms are modifications of a common code base. Even when the code bases are very similar, the features we extract add useful information for algorithm selection. For the scenarios where adding algorithm features decreases performance, we observe smaller changes. On SAT18-EXP, we tie for pairwise regression. For SAT11-RAND, the decrease is 1% in the gap closed for both regression and pairwise regression approaches.

For comparison, we use dummy algorithm features that allow us to assess to what extent the performance improvements we see are because we train a single model that can take the performance information of all algorithms into account or because algorithm features genuinely help to make the models more predictive. The results for performance predictions we presented in the previous section already strongly indicate that algorithm features genuinely help. Adding the dummy features decreases performance in the majority of cases. In most of the few cases where performance improves compared to only instance features, the differences are small – including our proposed algorithm features improves performance more often and by a larger amount. This conclusively demonstrates that algorithm features are useful for per-instance algorithm selection.

We see a similar trend in the results for the experiments when we consider PAR10 scores (Figure 2 in the appendix), where adding algorithm features improves performance for 6 out of 9 scenarios for both regression and pairwise regression approaches, with one tie. To determine statistical significance of our results, we used the Friedman test on the ranks achieved by every approach on each problem instance in a given scenario (total 45 cases), along with a Nemenyi post-hoc test. The difference in ranks between the baseline and our proposed approaches was statistically significant at 5% level for all scenarios, except in one case with regression models. The difference between instance features and our features was not statistically significant for the GRAPHS-2015 scenario. For pairwise regression models, the differences were statistically significant at the same level for all but five cases for the MAXSAT19-UCMS, SAT03-16_INDU, SAT11-HAND, SAT11-INDU, and TSP-LION2015 scenarios.

## 5.2 Algorithm Feature Importance

To provide additional insight into the usefulness of our algorithm features and to investigate the potential for further improvements, we conducted feature selection experiments for our algorithm features. Specifically, we performed forward selection without hyperparameter tuning to reduce experimental cost, following [Bischl et al., 2016]. We only considered our algorithm features for selection and always use the full set of instance features; we made this choice to keep the experimental effort manageable, and because results for feature selection experiments on instance features are available in the literature [Bischl et al., 2016], showing that a small number of instance features are sufficient to achieve good performance. We performed this analysis for both types of approaches; regression models and pairwise regression models.

The selected features for regression models are shown in Table 3, results for pairwise regression models are shown in Table 6 in the appendix. They are not consistent across scenarios. Features that measure the complexity of the algorithm in some way, such as average cyclomatic complexity, indentation, degree of AST nodes, and depth of AST leaves, are often chosen. Intuitively, it makes sense that they are predictive of the algorithm's performance – more complex code might indicate more sophisticated heuristics, but also more complex computation.

Tables 4 and 5 in the appendix show the percent gap closed between single and virtual best solvers for the full and reduced feature sets. Our results show that performance can improve when irrelevant features are removed, and even a very small set of features is often sufficient for obtaining good performance – in no case, more than five features are selected, and in many cases just a

| Scenario | Selected Features |
|---|---|
| GLUHACK-18 | average cyclomatic complexity |
| | variance of AST node degrees |
| | fraction of AST nodes that represent a type |
| GRAPHS-2015 | average maximum indent |
| MAXSAT19-UCMS | fraction of AST nodes that represent a statement |
| | fraction of AST edges that link a type node to a declaration node |
| SAT03-16_INDU | fraction of operators applied on `long long` |
| | fraction of operators applied on `float` |
| SAT11-HAND | entropy of the distribution of the degree of AST nodes |
| SAT11-INDU | average cyclomatic complexity |
| | fraction of AST edges that link a type node to a statement node |
| SAT11-RAND | average cyclomatic complexity |
| SAT18-EXP | total lines of code |
| TSP-LION2015 | entropy of the distribution of the depth of AST leaves |

Table 3: Feature sets chosen by forward selection for each scenario, optimizing PAR10 score for regression models.

| Scenario | Algorithm Features | Gap Closed (PAR10) | |
|---|---|---|---|
| | | Full Set | Reduced Set |
| GLUHACK-18 | 75 → 3 | 48 | 57 |
| GRAPHS-2015 | 75 → 1 | 53 | 72 |
| MAXSAT19-UCMS | 75 → 2 | 38 | 57 |
| SAT03-16_INDU | 75 → 2 | 46 | 53 |
| SAT11-HAND | 75 → 1 | 46 | 56 |
| SAT11-INDU | 75 → 2 | 27 | 50 |
| SAT11-RAND | 75 → 1 | 89 | 91 |
| SAT18-EXP | 75 → 1 | 59 | 66 |
| TSP-LION2015 | 75 → 1 | 26 | 12 |

Table 4: Algorithm feature selection results, showing number of selected features and percent gap between single and virtual best solver closed in terms of PAR10 for the full and reduced feature sets. We performed forward selection with regression on the entire set of code and AST features. Numbers were rounded to the nearest integer. Note that hyperparameters have not been optimized and results may be different to the algorithm selection results shown previously.

single one. This echoes similar findings for the number of instance features in [Bischl et al., 2016]. For TSP-LION2015, performance is worse with the reduced set of features for regression models. Adding a second feature to the one chosen previously does not improve performance and the feature selection process stops. There is likely a feature set that would lead to an increase in performance, but we were unable to exhaustively test all feature subsets because of the computational cost.

### 5.3 Discussion

We posit that the features we extracted from algorithms give useful information about those algorithms. However, the features are at a relatively low level that does not, for example, allow to conclude what type of approach or algorithm was used. There is no intrinsic reason that any of the features should be related directly to the performance of the algorithm on particular inputs, just

like there is no intrinsic reason that problem instance features should contain such information, for example, it is unclear why the number of connected components in a constraint graph tells us anything about how difficult the problem instance is to solve. We do see that algorithm selection performance improves, which in itself provides a motivation for our approach, again similar to how algorithm selection in general is motivated by the success of using instance features. Of course, our approach (like algorithm selection and, indeed, automated machine learning) does not always work, but we do show its promise through our experimental results.

## 6 Conclusions and Future Work

We presented, to the best of our knowledge, the first approach that uses features characterizing software for algorithm selection. These features can be extracted automatically, using off-the-shelf analysis software – as long as the source code for the given set of algorithms is available, our approach can be applied. We hence retain one of the main advantages of current black-box meta-algorithmic techniques – their broad availability. The results from our experiments clearly demonstrated that using these additional features can yield significant performance improvements for algorithm selection.

We also showed that only a small subset of the rich set of algorithm features we considered is required for achieving these improvements. While our approach does not improve performance for all scenarios we considered in our study, we achieved improvements on average across a wide variety of scenarios. These results demonstrate the promise of our approach and the potential for algorithm features to improve algorithm selection and other meta-algorithmic techniques.

Our work can be extended in various directions. It would be interesting to consider scenarios where the performance metric of interest is the solution quality rather than runtime such as in machine learning. This could potentially extend our approach into the domain of automated machine learning.

Furthermore, the type of software features we consider here is mostly limited to algorithm selection – for parameter tuning and algorithm configuration, where the difference a parameter makes is often not directly reflected in the code base and becomes only apparent at execution time, different software analysis techniques will be required to effectively distinguish between parameter configurations. While this requires significant further work, the ability to predict the performance of a parameter configuration more accurately has very substantial potential for impact across many areas of AI.

Finally, a more in-depth analysis of the relationship between software features, instance features, and the performance of a given algorithm on a given problem instance could provide insight into how the novel features we propose here facilitate improved algorithm selection.

### 6.1 Acknowlegments

Damir Pulatov and Lars Kotthoff are supported by NSF grant #1813537.

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

## A  Algorithm Selection Scenarios

We require the source code for the algorithms to be available, and hence limited our empirical study to ASlib scenarios for which we were able to obtain source code. As this ruled out quite a few of the benchmarks, we created additional scenarios for our evaluation and contributed them back to ASlib. We focus on scenarios that predict the runtime of algorithms in this paper; the methodology extends to scenarios with other performance measures as well though.

We performed experiments on six pre-existing ASlib scenarios – TSP-LION2015, SAT11-RAND, SAT11-HAND, SAT11-INDU, SAT03-16_INDU, and GRAPHS-2015. These scenarios represent a variety of different applications of algorithm selection.

For four ASlib scenarios, we had to exclude some of the solvers. In particular, for SAT11-RAND, we excluded sattime2011_2011.03.02, because we were unable to identify the source code and version corresponding to the scenario; for SAT11-HAND, we excluded jMiniSat_2011, sattime._2011.03.02, and sattime_2011.03.02 for the same reason. We also excluded Sol_2011.04.04, as our algorithm analysis tools currently do not support C# code. For the SAT03-16_INDU scenario, we excluded cominisatps and riss5, again because we were unable to identify the exact version of the source code used for the scenario.

The Glasgow algorithms from the GRAPHS-2015 scenario were four parameterizations of the same underlying algorithm. Therefore, we excluded three versions of Glasgow from the scenario.

## B  Algorithm Selection

Due to the size of the scenario and the amount of computation required, we were unable to consider all algorithms for SAT18-EXP when using models that predict performance difference (666 models would be required for the full scenario to compare to existing approaches; our proposed approach does not suffer from this limitation and we omit solvers only to be able to compare to other approaches). We removed seven solvers that were the best on fewer than six out of 343 problem instances: Candy, Minisat.v2.2.0.106.ge2dd095, Lingeling, Riss7.1.fix, Sparrow2Riss.2018.fixfix, glucose3.0, and expGlucose. The first two of these were never the best for any of the instances. This leaves us with 30 algorithms in this scenario for 435 pairwise performance models; the overall performance did not change significantly. The mean MCP for the single best solver (the individual solver that performs best on average across all problem instances) decreased from the original 1220.06 to 1211.12 CPU seconds, and the mean PAR10 score for the virtual best solver (the oracle that for each problem instance determines, without error, the best algorithm for solving it) increased from the original 9841.23 to 9850.17 CPU seconds. The PAR10 score for the single best solver and MCP for the VBS remained the same. All other scenarios were used without modifications, and the full SAT18-EXP scenario was used for experiments where the performance of each algorithm is predicted directly and a selection is made based on this predicted performance.

## C  Results

When considering results in terms of PAR10 scores, we see the largest performance improvements for TSP-LION2015, where we close additional 71% and 48% of the gap for regression and pairwise regression models, respectively. For SAT11-INDU, we close 19% more of the gap for both regression and pairwise regression models; 3% and 18% more compared to dummy features. For MAXSAT19-UCMS, we close additional 10% and 8% gap for regression and pairwise regression, respectively.

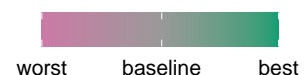

## Regression     Pairwise Regression

| | Regression | | | | | Pairwise Regression | | | | |
|---|---|---|---|---|---|---|---|---|---|---|
| | I | II | IC | IA | ICA | I | II | IC | IA | ICA |
| TSP–LION2015 | −32 ±173.82 | −75 ±214.36 | −30 ±250.38 | **39** **±36.11** | 26 ±27.98 | −48 ±157.04 | −48 ±264.32 | −29 ±176.13 | −1 ±9.14 | **0** **±4.74** |
| SAT18–EXP | **60** **±23.21** | 53 ±26.18 | 55 ±17.8 | 59 ±28.2 | 59 ±23.37 | **56** **±15.64** | 27 ±22.44 | **56** **±8.44** | 45 ±19.55 | 42 ±21.52 |
| SAT11–RAND* | **89** **±8.05** | 88 ±10.26 | 86 ±12.02 | 88 ±10.43 | **89** **±10.14** | **90** **±7.64** | **90** **±7.64** | 89 ±6.73 | 80 ±11.94 | 81 ±10.66 |
| SAT11–INDU | 11 ±697.64 | 27 ±692.16 | **30** **±346.28** | 27 ±336.48 | 27 ±339.99 | 28 ±336.2 | 29 ±345.57 | **47** **±355.73** | 37 ±339.11 | 27 ±337.75 |
| SAT11–HAND* | 39 ±117.28 | 43 ±141.18 | 39 ±117.23 | 45 ±111.06 | **46** **±140.42** | 43 ±138.23 | 39 ±232.4 | 39 ±218.74 | **52** **±53.68** | 50 ±47.33 |
| SAT03–16_INDU* | 45 ±19.4 | **48** **±15.07** | 47 ±19.26 | 46 ±16.47 | 46 ±25.52 | 48 ±22.18 | 36 ±24.47 | **51** **±17.85** | 35 ±17 | 41 ±11.26 |
| MAXSAT19–UCMS | 40 ±23.23 | 38 ±44.94 | **50** **±28.08** | 41 ±31.09 | 38 ±35.34 | 46 ±39.54 | 50 ±32.36 | **54** **±42.43** | 21 ±35.83 | 25 ±43.08 |
| GRAPHS–2015* | 53 ±946.93 | 51 ±969.43 | **66** **±973.37** | 57 ±977.17 | 53 ±2002.54 | 61 ±1022.89 | **67** **±1997.84** | 57 ±970.8 | 66 ±979.19 | 54 ±964.25 |
| GLUHACK–18 | 43 ±40.95 | 43 ±68.31 | **52** **±33.3** | 41 ±71.25 | 48 ±46.69 | 42 ±43.06 | 43 ±46.94 | **46** **±22.52** | 39 ±50.46 | **46** **±23.38** |

Figure 2: Impact of algorithm features on the performance of algorithm selection based on regression and pairwise regression models. The top value in each cell is the average percentage value of the gap closed between single best (0) and virtual best solver (100) in in terms of PAR10 scores across cross-validation folds; the bottom value is the standard deviation. Values greater than zero indicate that performance is better than the single best solver, negative numbers indicate that it is worse. Average values have been rounded to integers and standard deviations to two decimal places. Starred ASlib scenarios were modified to exclude some algorithms. The column on the left shows the baselines to which we compare – instance features only (I) and dummy ID features (II). The algorithm feature sets we consider are instance and code features (IC), instance and AST features (IA), and instance, code, and AST features (ICA). The best value for a particular scenario is shown in **bold**.

| Scenario | Algorithm Features | Gap Closed (PAR10) | |
|---|---|---|---|
| | | Full Set | Reduced Set |
| GLUHACK-18 | 75 → 3 | 46 | 57 |
| GRAPHS-2015 | 75 → 1 | 54 | 72 |
| MAXSAT19-UCMS | 75 → 2 | 25 | 57 |
| SAT03-16_INDU | 75 → 2 | 41 | 53 |
| SAT11-HAND | 75 → 1 | 50 | 56 |
| SAT11-INDU | 75 → 2 | 27 | 50 |
| SAT11-RAND | 75 → 1 | 81 | 91 |
| SAT18-EXP | 75 → 5 | 42 | 63 |
| TSP-LION2015 | 75 → 1 | 0 | 12 |

Table 5: Algorithm feature selection results, showing number of selected features and percent gap between single and virtual best solver closed in terms of PAR10 for the full and reduced feature sets. We performed forward selection with pair regression on the entire set of code and AST features. Numbers rounded to integers. Note that hyperparameters are not tuned and results may be different to the algorithm selection results shown previously.

| Scenario | Selected Features |
|---|---|
| GLUHACK-18 | average cyclomatic complexity |
| | variance of AST node degrees |
| | fraction of AST nodes that represent a type |
| GRAPHS-2015 | average maximum indent |
| MAXSAT19-UCMS | fraction of AST nodes that represent a statement |
| | fraction of AST edges that link a type node to a declaration node |
| SAT03-16_INDU | fraction of operators applied on `long long` |
| | fraction of operators applied on `float` |
| SAT11-HAND | entropy of the distribution of the degree of AST nodes |
| SAT11-INDU | average cyclomatic complexity |
| | fraction of AST edges that link a type node to a statement node |
| SAT11-RAND | average cyclomatic complexity |
| SAT18-EXP | entropy of the distribution of the degree of AST nodes |
| | mean of the AST clustering coefficient |
| | variance of the AST clustering coefficient |
| | maximum depth of AST leaves |
| | fraction of AST nodes that represent a literal |
| TSP-LION2015 | entropy of the distribution of the depth of AST leaves |

Table 6: Feature sets chosen by forward selection for each scenario, optimizing PAR10 score for predicting the performance difference between two algorithms.

