# OpenReview forum: "Opening the Black Box: Automated Software Analysis for Algorithm Selection"
_automl.cc/AutoML/2022/Track/Main — AutoML-Conf 2022 (Main Track)_

### Official Review · Reviewer_oJUM · 2022-03-31

**Potential Impact On The Field Of Automl Rating:** 4
**Technical Quality And Correctness Rating:** 4
**Clarity Rating:** 4

**Summary Of Contributions:**

Automatically selecting an algorithm to optimally solve a set of problems is known as the per-instance algorithm selection problem.  This meta-optimization problem has been especially studied in the context of optimization problems (e.g., SAT solving). Important improvements of state-of-the-art solving techniques have been achieved in both discrete and continuous optimization domains. Most of the time, the general methodology consists in using features that are associated with the problem instances and to predict the performances of a portfolio of algorithms (that may consist of different algorithms or different settings of an algorithm).

In this paper, the authors propose to use features that describe the solving algorithms. It means that some kind of knowledge is required while previous methods claimed that they considered black-box solving techniques. Nevertheless, the authors propose general code-based features that can be easily extracted from the available softwares. They then perform an extensive set of experiments on established benchmarks that highlight the benefit of their proposed approach.

**Clarity:**

The paper is well written and organized. A description of state-of-the-art existing methods allows even non-expert readers to understand the motivations of the work and its novelty. The experimental process and the resulting evaluation are also precisely and clearly defined.

**Overall Review:**

Pro :

- The originality of the proposed features extraction and use from solvers’ code

- Fair and well-chosen experimental process

- Obtained performances that assess the benefits of the approach

- Quality of the writing

- The approach could be extended to other algorithm selection problems (e.g. in ML)

Cons:

- The proposed approach is not suitable for parameter tuning of solvers (which is a possible target of algorithm selection methods

- The explanation of the relevance of the solvers’ features could be difficult to produce



**Potential Impact On The Field Of Automl:**

The approach is originally supported by generic code analysis tools that allow the authors to keep the generality of their proposed approach. The authors focus on optimization problem solving but similar techniques could certainly be used on other algorithm selection pipelines.

**Reproducibility:**

As mentioned above, reproducibility is fully addressed in this paper and the reproducibility checklist is precisely filled.


**Review Confidence:**

4: You are confident in your assessment, but not absolutely certain. It is unlikely, but not impossible, that you did not understand some parts of the submission or that you are unfamiliar with some pieces of related work.

**Review Rating:**

5: Accept, good paper

**Review Summary:**

This is an interesting work whose motivations are clearly stated. The contribution is relevant with state-of-the-art techniques. The method can be adapted to other domains. Nevertheless, it already provides interesting results for discrete optimization solving. The experiments are based on a significant set of benchmarks. The presentation of the experimental results is very satisfactory.


**Technical Quality And Correctness:**

The main motivations and tools used in this paper are clearly described. The state of the art is rather complete. The experiments are precisely described and therefore the fairness and reproducibility of the performed experimental study seem  to be fulfilled.

---

### Official Review · Reviewer_FPW5 · 2022-04-01

**Potential Impact On The Field Of Automl Rating:** 4
**Technical Quality And Correctness Rating:** 3
**Clarity Rating:** 3

**Summary Of Contributions:**

The paper proposes to use features based on the algorithm's implementation (=source code) in automated algorithm portfolio selection.
The authors use standard software analysis features such as abstract syntax tree (AST) to extract it from the algorithms. They use standard algorithm portfolios and two standard tasks and evaluate their approach on a standard library ASPLib.

**Clarity:**

The paper is written clearly.
Somehow I missed some detail on which exact approach for portfolio selection was used but it does not really play a major role.

**Overall Review:**

I do appreciate the idea to consider algorithm features and that is certainly a positive point about this paper.
However, how useful are the chosen features? What do they say about the actual functionality of the algorithm?
What if exactly the same algorithm is implemented in some way that changes the features but not its functionality?

To me I fail to see why and how such features would help with algorithm selection - well now you obviously can argue that the results are good and yes they are certainly encouraging. Are the baselines good enough?  For instance the name-based embedding dummy variable I mentioned above - would that make a difference? I do not want to sound too negative but I just cannot see the relation.

To me it is more a conceptual issue. If you would have said we analyzed the paper or github history for each solver and extracted the main functionality (like used 'binary resolution' or had 'XOR' support or the relationship - this is cryptominisat 2.0 which comes from cryptominisat 1.0 and is a derivative of MiniSat or look at the parameters documentation) and used those features to do algorithm selection I would argue that would be a lot more meaningful.

The argument that this is great since one can build exactly one model for algorithm portfolios is true but nothing special either. You do cite a lot of work, but for instance not consider work such as "Probabilistic Matrix Factorization for Automated Machine Learning
Nicolo Fusi, Rishit Sheth, Huseyn Melih Elibol, NeurIPS, 2018.
I mention this one explicitly since it not only build a single model, but is actually a suitable match for your setup since it is essentially uses a recommender system. Why is that relevant? Just consider the Netflix setup - here movies are algorithms and users are problem inputs. This seems like one could do an interesting analysis (like in Netflix) what kind of algorithms like which problems - which has also been previously done. However, it would help if the features that characterize the algorithms are actually somewhat meaningful.


**Potential Impact On The Field Of Automl:**

In general, novel ways of Improving algorithm portfolios will always have an important impact on AI.
However, I doubt that the approach presented here will have that impact.

**Reproducibility:**

Seems mostly reproducible to me although there is quite some machinery to setup to extract the features from the code and for some it did not work etc. Not always easy to reproduce all of this but not really the fault of the authors either.
Overall it is good direction, but I wish the authors had chosen features that actually capture some functionality.

**Review Confidence:**

5: You are absolutely certain about your assessment. You are very familiar with the related work and checked all the details carefully.

**Review Rating:**

3: Marginally below the acceptance threshold (use sparsely)

**Review Summary:**

I do like the overall idea of using algorithm features, but I do not see how the chosen features provide a meaningful characterization of the algorithm for algorithm selection. One can argue that the the results seem reasonable modulo some more sophisticated 'dummy features', but to me it definitively does not add any meaningful transparency to the black-box algorithm.

**Technical Quality And Correctness:**

Seems technically correct to me.
One question I had is on the 'dummy solver id'.
In SAT solver competition it is often the case that the same solvers but different versions are used. Like Zchaff and Zchaff_200x.
Would it not be a better "dummy ID" to use the actual solver names and use some embedding (like word2vec or USE) as features?

---

### Official Review · Reviewer_JxKZ · 2022-04-04

**Potential Impact On The Field Of Automl Rating:** 4
**Technical Quality And Correctness Rating:** 4
**Clarity Rating:** 4

**Summary Of Contributions:**

This paper introduces an approach to allow an in-depth analysis of the algorithm selector. It proceeds by investigating the underlying surrogate model. Concretely, the authors proposed using new features to describe algorithms. The latter algorithm features combined with instance features are fed to a surrogate model to predict the most promising algorithm. Finally, the paper presents an empirical validation of the approach on ASlib, showing the merits of the added algorithm features while yielding new insights.


**Clarity:**

The paper is well written and easy to follow in general. A minor recommendation for the authors is to add a formal definition of single and virtual best solvers. They are not evident from my perspective.

**Overall Review:**

The contribution of this paper is technically novel and solid. Moreover, the proposed approach yields superior performance than the considered baseline (algorithm selectors without algorithm features).
The paper also provides insight into the relative gain of each feature set (instance, code, and AST), suggesting the merits of the algorithm features. Finally, further analysis is also presented to investigate the importance of algorithm features.

Beyond the importance of features, I wonder if the authors did observe patterns that characterize the relationship between instance features and features of the best algorithm. In my opinion, such an insight provides the underlying decision being made by the selector.

**Potential Impact On The Field Of Automl:**

This paper directly addresses the algorithm selection problem. In addition, it also provides an interesting approach to analyzing algorithm selectors.

**Reproducibility:**

According to the authors, this contribution has been merged to ASlib; thus, I deduce that the approach is fully reproducible.

**Review Confidence:**

3: You are fairly confident in your assessment. It is possible that you did not understand some parts of the submission or that you are unfamiliar with some pieces of related work.

**Review Rating:**

5: Accept, good paper

**Review Summary:**

From my perspective, this paper presents solid contributions both technically (analyzing algorithm features) and empirically (yielding superior performances). For this reason, I recommend acceptance.

Nevertheless, I am not an expert in the Algorithm Selection domain and thus can not situate/assess the paper w.r.t the current state-of-the-art algorithm selectors, justifying my review confidence score set to 3.

**Technical Quality And Correctness:**

The contribution of this paper is technically sound. To my best knowledge, the approach to adding algorithm features used inside the algorithm selector is new. I believe the results and analyses presented in this paper will be useful to a large community of researchers.

---

### Official Review · Reviewer_Mo3K · 2022-04-05

**Potential Impact On The Field Of Automl Rating:** 2
**Technical Quality And Correctness Rating:** 3
**Clarity Rating:** 4

**Summary Of Contributions:**

The paper proposes to select algorithms based, not only on instance features (as is classically the case) but also based on algorithm features that are derived from the algorithm’s source code using program analysis. Given these features, they then selects the algorithm from the portfolio that is predicted to perform best according to a single model of algorithm performance across instance and algorithm feature space. They validate the usefulness of these program features empirically on ASlib and further analyze feature importance.

**Clarity:**

The paper is well-written, there is a clear line of argument throughout, making it easy to follow.

**Overall Review:**

Reasons to accept:
- Interesting/novel premise (despite prior art): The paper proposes a general extension to the algorithm selection framework, extending it with algorithm features, exploiting the fact that algorithms are often not truly a black box, yet are treated as such. This proposed framework is interesting, and despite prior art pursuing this line of work exists (see technical quality), it is to the best of my knowledge the first method using features derived from program analysis. Furthermore, in meta-algorithmics in general this white box approach is relatively unexplored.
- Clear representation: The paper is well-written, there is a clear line of argument throughout, making it easy to follow.

Reasons to reject:
- Limited relevance to AutoML (see Impact on AutoML)
- Not entirely convinced of usefulness features: It is not obvious to me why the algorithm features (mostly complexity measure) considered in the paper would be useful. Except for l260 the paper provides little motivation of why these features would be useful. While the experimental results do provide some evidence that these features are useful, I do not agree that these are conclusive (l235).
- Failing to provide code (see reproducibility)


**Potential Impact On The Field Of Automl:**

The paper is a fundamental algorithm selection paper and does not directly apply to AutoML, e.g., in none of the experiments machine learning was the target problem (the as lib scenarios considered are classical OR problems). While it is definitely true that meta-algorithmics drive many AutoML systems, this is mainly true for algorithm configuration and lesser for selection. Arguably this is due to the lack of good instance features. Augmenting these instance features with program features (as suggested by the authors) may help, however it is unclear whether the specific program complexity features would be useful. More promising arguably is the approach of using execution time features the authors suggest at the end (l289). However, there already exists various previous work in this direction that open up the ML black box, e.g., grey box HPO approaches like HyperBand (https://arxiv.org/abs/1603.06560), and work on learning curve prediction (e.g., https://ml.informatik.uni-freiburg.de/wp-content/uploads/papers/17-ICLR-LCNet.pdf)

I would therefore envision the impact of this work on AutoML (specifically) to be minimal, however, beyond ML, the proposed extension of AS is interesting / more novel and could have a wider impact.


**Reproducibility:**

The authors did not provide code alongside their submission to allow reproduction. I do not entirely follow the argument: “We have already modified algorithm selection libraries we references (llama and aslib) with our approach. Please do not check the libraries to ensure Anonymity.”
While I agree that one should not break double-blindness (and linking aslib would do that), I do not see why the same code could not have been provided alongside the submission in an anonymized form.


**Review Confidence:**

4: You are confident in your assessment, but not absolutely certain. It is unlikely, but not impossible, that you did not understand some parts of the submission or that you are unfamiliar with some pieces of related work.

**Review Rating:**

4: Marginally above the acceptance threshold (use sparsely)

**Review Summary:**

This paper has an interesting premise, and the authors present their study thereof in a clear way. Therefore, I feel that this paper, despite its shortcomings (limited relevance AutoML, non-conclusive results, missing code), would make a fine contribution to the conference, assuming the issues raised (related work, softening some claims) are addressed.

**Technical Quality And Correctness:**

I found this work to be mostly sound, however there are a few concerns that need addressing and that weaken some of the claims made:

Generality of the proposed approach: The abstract states ‘without sacrificing the universal applicability of meta-algorithmic techniques (l11).  However, in the experiments the authors excluded algorithms from selection because there source code was not available (l136). The approach assumes source code to be available, and in algorithm selection that is not always the case. This can either be resolved by weakening the claim (e.g., as in l274) or treating all features as being missing (except for the dummy ID feature) and including these in the experiments.

Related work: The approach of using features from static program analysis is novel and interesting (main novelty claim as stated l273), nonetheless there is various related work considering other ‘algorithm features’ in meta-algorithmics that is not referenced, and at some points (e.g., l110) the authors seem not aware that such work exists. The most closely related work I am aware of:
- https://www.ijcai.org/Proceedings/16/Papers/085.pdf
Made a general/formal argument for ‘opening up the black box’ in meta-algorithmics. While the focus of this work was on runtime information and its application to algorithm configuration / dynamic algorithm selection, the article suggests that static program analysis could also be useful.
(Relates to lines: L39/L88/L291)
- https://link.springer.com/chapter/10.1007/978-3-030-61527-7_21
Does algorithm selection for machine learning based on instance and algorithm features.
- Work on per-instance algorithm configuration (PIAC) can be viewed to represent the algorithm by its hyperparameters, and surrogate models for Algorithm Configuration (https://www.researchgate.net/publication/252075395_Performance_Prediction_and_Automated_Tuning_of_Randomized_and_Parametric_Algorithms_An_Initial_Investigation, https://link.springer.com/article/10.1007/s10994-017-5683-z) as modelling the cost conditional on instance features and hyperparameters

---

### Official Review · Reviewer_s7pd · 2022-04-11

**Potential Impact On The Field Of Automl:** N/A for reproducibility reviewers
**Potential Impact On The Field Of Automl Rating:** 3
**Technical Quality And Correctness:** N/A for reproducibility reviewers
**Technical Quality And Correctness Rating:** 3
**Clarity:** N/A for reproducibility reviewers
**Clarity Rating:** 3

**Summary Of Contributions:**

N/A for reproducibility reviewers

**Overall Review:**

N/A for reproducibility reviewers

**Reproducibility:**

The authors did not provide code, data, or instructions for reproducibility.

**Review Confidence:**

4: You are confident in your assessment, but not absolutely certain. It is unlikely, but not impossible, that you did not understand some parts of the submission or that you are unfamiliar with some pieces of related work.

**Review Rating:**

4: Marginally above the acceptance threshold (use sparsely)

**Review Summary:**

N/A for reproducibility reviewers

---

### Meta-Review · Area_Chair_jtLk · 2022-05-06

**Recommendation:** Accept
**Confidence:** 4

**Metareview:**

Technical quality: Sound, no major issues

Clarity: Paper is clear and easy to read. However, lacks real efforts to understand why the chosen features work

Originality: Obtaining features from program analysis of the algorithm source code is original

Significance: While the proposed approach improves over baselines (mostly an ablation study), this is a bit of a weak point, since no real attempt is undertaken to understand why such features would be useful.

This paper adresses automatic algorithm selection from a portfolio, in order to best solve a problem instance. The main novelty is the use of algorithm features extracted from the source code by program analysis (e.g., features derived from the AST). The authors present an evaluation on a number of standard benchmarks of problem instances, where it is shown that the new algorithm features improve performance in most cases, compared to only using instance features and an algorithm indicator.

Reviewers agree that the use of features extracted from program analysis is interesting and original, given that algorithm source code is often available and standard tools can be used to extract them. Also, the empirical evaluation is deemed sound, even though some more baselines could have been compared against (e.g., simple algorithm features used in previous work).

The paper lacks a motivation why the extracted algorithm features would be useful, apart from the empirical evidence (which could have been stronger). The feature importance study by forward selection closely follows prior work (Bischl etal, 2016) and remains inconclusive. More effort from the authors to tease out why these features works, would have made the work a lot stronger. As it stands, it remains a purely empirical observation. It is therefore recommended the authors tone down statements like "opening the black box", or at least clearly explain what they mean.

The authors are strongly encouraged to publish their code, so others can explore these type of algorithm features as well.

---

### Decision · Program_Chairs · 2022-05-13

Accept